# Research on the Effect of Evidence-Based Intervention on Improving Students’ Mental Health Literacy Led by Ordinary Teachers: A Meta-Analysis

**DOI:** 10.3390/ijerph20020949

**Published:** 2023-01-04

**Authors:** Yuanyuan Liao, Moses Agyemang Ameyaw, Chen Liang, Weijian Li

**Affiliations:** College of Teacher Education, Zhejiang Normal University, Jinhua 321004, China

**Keywords:** evidence-based, students’ mental health literacy, ordinary teachers, meta-analysis

## Abstract

Background: the purpose of this study was to systematically review the effects of intervention experiments led by ordinary teachers to improve students’ mental health literacy and to provide evidence-based research and new ideas for improving students’ mental health literacy. Methods: A systematic search using 5 English (Web of Science, PubMed, ProQuest, EBSCO, Springer Link) and 3 Chinese (CNKI, WanFang, and VIP) databases was initiated to identify controlled trials assessing the immediate effect and delay effect of the intervention experiment led by ordinary teachers on improving students’ mental health knowledge, anti-stigma, willingness, or behavior to seek-help. Results: a total of 14 experiments with 7873 subjects were included. The results showed that the immediate effect of the intervention on promoting students’ mental health knowledge [g = 0.622, 95% CI (0.395, 0.849)] and anti-stigma [g = 0.262, 95% CI (0.170, 0.354)] was significant, but the amount of delay effect is not significant. Conclusions: the results of this review show that ordinary classroom teachers can effectively participate in projects to improve students’ mental health literacy, significantly improve students’ mental health knowledge and attitudes towards psychological problems, and make up for the shortage of full-time mental health teachers in schools. In future, more attention should be paid to students’ mental health literacy, and evidence-based intervention research should be strengthened. Furthermore, we can improve students’ mental health literacy and avoid poor mental health by addressing delays in early intervention, as well as improve experimental design, prolong the intervention time, and improve the effectiveness of the intervention.

## 1. Introduction

The mental health of students is a major issue related to personal and national development. According to the statistics of the World Health Organization (WHO), nearly 20% of the world’s adolescents suffer from different degrees of psychological disorders, which has become a global challenge and an important strategy and priority for the development of public health in various countries [1,2,3] (Kessler et al., 2000; WHO, 2001; WHO, 2000–2011). The initial onset of most mental health problems occurs before the age of 25 [1,4,5] (Kessler et al., 2000; Kelly CM, Jorm AF and Wright A, 2017; Rusch et al., 2011), and the degree of mental health problems is usually mild to moderate. The response to the measures is positive, and the treatment effect is good [6,7,8] (Kessler RC, Avenevoli and Costello J, 2012; Kutcher, 2011; Rutter et al., 2010), if timely identification and intervention during this period can significantly improve the individual Attitudes, behaviors, and mental health levels of seeking professional help [4] (Kelly CM, Jorm AF and Wright A, 2017). However, surveys have shown that 70–80% of adolescents with mental illnesses do not receive the mental health services they need, especially in developing countries [9,10] (Ren Zhihong et al., 2020; Thornicroft, G., 2007). The main reason is the delay of early intervention caused by an insufficient number of full-time and part-time teachers for mental health in schools, low levels of specialization, limited mental health service resources, and low mental health literacy in students [3,10,11,12,13] (WHO, 2011; Thornicroft, 2007; Patel et al., 2007; Tolan and Dodge, 2005; Waddell et al., 2005).

Mental health literacy is the knowledge, beliefs, and behaviors about mental disorders, including understanding how to achieve and maintain positive mental health, understanding mental disorders and their treatment, reducing the stigma associated with mental disorders, and improving self-help and helping others [14,15,16,17] (Jorm et al., 1997; Jorm, 2012; Kutcher, Bagnell and Wei Y, 2015; Kajawu et al., 2016). At present, there are two main ways to measure mental health literacy: case interviews and questionnaire surveys [9] (Ren zhihong et al., 2020). Jorm et al. (1997) used case interviews to investigate public awareness of the causes and risk factors of depression and schizophrenia [14]. There are various forms of questionnaires, but no questionnaire has been widely used at present [9] (Ren zhihong et al., 2020). Improving mental health literacy has been widely recognized by countries and international organizations as the key to promoting individual mental health. Due to a lack of mental health knowledge, limited ability to identify mental disorders, and the impact of mental illness stigma, students are less willing to seek formal help, so they are more dependent on others for help to identify mental illness symptoms and guide them to appropriate interventions, such as friends, family, teachers, etc. Among them, teachers, as the adults with the most contact with students’ campus life, have unique advantages in identifying, helping, and supporting students’ mental health [18,19,20] (Atkins et al., 2011; McGorry et al., 2011; Rowling, 2015).

In the past two decades, many countries have carried out evidence-based intervention projects based on the participation of ordinary schoolteachers and focusing on improving students’ mental health literacy (MHL) as an important strategy and approach to identifying early symptoms in adolescents, reducing stigma, and improving the effectiveness of help-seeking [21]. These projects included the Adolescent Mental Health First Aid Program (United States) [22] (Theda Rose et al., 2017), Mental Health Teaching Program (United Kingdom) [23] (Paul B et al., 2009), National Curriculum for Personal Development, Health and Physical Education (Australia) [24] (Yael Perry et al., 2014), Middle School Students’ Knowledge, Attitudes and Help for Depression (Hong Kong, China) [25] (Eliza s et al., 2016). Teachers use videos in the classroom to let students contact mental health patients, teach mental health knowledge and classroom seminars, and achieve other means to increase students’ mental health knowledge, reduce stigma, and promote a help-seeking willingness and actual help-seeking behavior. However, the experimental results are different. For example, in the intervention of mental health knowledge, Stan Kutcher (2015) and Alan Mcluckie (2014) [16,26] measured immediately after the experiment and showed that school mental health knowledge improved to a significantly large effect size (*p* < 0.001, d ≥ 0.8). However, Amanda J. Nguyen (2020) showed a significantly small effect size (*p* < 0.001, d ≤ 0.2) immediately after the students’ mental health knowledge intervention trial [27], making it difficult for us to accurately grasp whether the mental health courses provided by ordinary teachers can effectively improve the students’ mental health literacy and the improvement effect. In addition, the existing meta-analysis has some limitations. For example, Yifeng Wei (2013) et al. systematically reviewed a total of 27 school-based mental health literacy programs with participants aged 12–25 [28], but this meta-analysis was not aimed at teacher-led projects. Ordinary teachers refer to teachers who are not full-time mental health teachers or part-time mental health teachers in schools. The intervention project led by ordinary teachers can make all teachers participate in school mental health education projects and effectively alleviate the shortage of teachers in school mental health education.

The purpose of this study is to explore the immediate effect and delay effect of the intervention project for improving students’ mental health literacy, led by ordinary schoolteachers, on students’ mental health knowledge, anti-stigma awareness, and help-seeking behavior.

## 2. Materials and Methods

This meta-analysis was conducted in accordance with the Preferred Reporting Items for Systematic Reviews and Meta-Analyses (PRISMA) guideline [29] (Moher et al., 2009).

### 2.1. Search Strategy

Total 2 reviewers independently searched the literature using the following English and Chinese databases: Web of Science (all years), PubMed (all years), ProQuest (all years), EBSCO (all years), Springer Link (all years), the Chinese National Knowledge Infrastructure (CNKI, all years), Wanfang (all years), and the Chinese Scientific Journal (VIP, all years). The searches were conducted from inception through October 2021. Discrepancies between the 2 reviewers (LYY, LC) were discussed until a consensus was reached. Any disagreements regarding the inclusion were discussed and resolved with the third reviewer (LWJ). The search terms used in this study were based on a previous related meta-analysis: school-based, teacher participate in, teacher led, student, mental health literacy, attitude, stigma, knowledge, help-seeking, seek care, experimental intervention, mental health education, and mental health curriculum. Chinese translations of these terms were used in Chinese databases. 

### 2.2. Inclusion and Exclusion Criteria

Literature meeting the following criteria were included in the meta-analysis: (1) experimental intervention studies based on schools that promote students’ mental health literacy knowledge, attitudes (stigma), and help-seeking, or one of these; (2) general classroom teacher-led interventions (school full-time or part-time mental health teachers, mental health specialists, or professionals from other institutions were excluded); (3) the research object was normal subjects, excluding natural disasters (such as earthquakes), wars, natural diseases; (4) the included research must be a peer-reviewed journal paper or dissertation; (5) the experiment must have pre-test, post-test or pre-test, or post-test and follow-up measurements; (6) experimental data report is complete and must contain sample size, mean (M), standard deviation (sd), or the independent sample t-test value or effect size d used to calculate the overall effect size of the intervention.

### 2.3. Study Selection, Data Extraction and Coding

A total of 2 reviewers (LYY, LC) independently screened studies based on title, abstract, and full text. Discrepancies were discussed until a consensus was reached, and any disagreements regarding inclusion were discussed and resolved with a third reviewer (LWJ). 2 reviewers extracted and summarized the following relevant data from all original articles: (1) basic characteristics of the included studies (i.e., author, country, date of publication, type of experiment, time of data measurement, type of data measurement, teacher teaching method, measurement questionnaire); (2) basic characteristics of the participants (i.e., school period, sample size, duration of intervention); (3) outcome parameters.

In order to prevent the influence of selection bias on the meta-analysis results in the data extraction and coding process, the study adopted the method of simultaneous coding by two researchers (Table 1).

### 2.4. Publication Bias and Sensitivity Testing

A funnel plot combined with Egger’s linear regression was used to test whether the original studies included in the meta-analysis had publication bias. If the effect values are evenly distributed around the top of the inverted funnel in the funnel chart, it means that there is less possibility of publication bias from the perspective of subjective judgment; if the *p*-value in Egger’s linear regression is not significant (*p* < 0.001), it means that it is an objective explanation; there is no publication bias. The stability of the meta-analysis results was tested by over-sensitivity analysis, that is, the effect value of each original study was removed separately and then the combined effect size was calculated again. If the combined effect size did not change significantly, the meta-analysis results were relatively stable [30] (Morgan, Ross, and Reavley, 2018).

### 2.5. Data Analysis 

#### 2.5.1. Combined Effect Size Calculation

Taking students’ mental health literacy intervention knowledge and the immediate effect and delay effect of stigma and help-seeking as outcome variables, the main effect test was conducted to investigate the effect of students’ mental health literacy intervention projects led by ordinary teachers on students’ mental health knowledge and stigma. The effect of intervention on attitudes and willingness to seek help or behavior. The standardized mean difference Hedge’s g (corrected by Cohen’s d) was used as the combined effect size of the intervention. If the study did not report the mean and standard deviation, the independent sample t-test value or the effect size d value was extracted, and then the overall effect value obtained by the CMA3.0 (comprehensive mate-analysis 3.0) software was input. The evaluation standard of the effect size is that the absolute value of the combined effect size d ≤ 0.2 is a small effect size, 0.79 ≥ d ≥ 0.21 is a medium effect size, and d ≥ 0.8 is a large effect size [31] (Cohen, 1988).

#### 2.5.2. Model Selection and Heterogeneity Testing

The random effect model is intended to be used for the calculation of the effect size, which is mainly based on the following three points: (1) the random effect model in the meta-analysis assumes that each independent effect value is based on a set of multiple true effect values, so there is a certain amount of independent effect values. However, the result data differs from multiple independent studies [9,32] (Ren Zhihong et al., 2020; Yamaguchi, S., 2018). (2) The results of this study can be generalized to other contexts to a certain extent [33] (Carrero, Vila, and Redondo, 2019). (3) Random-effects models enable wider confidence intervals for pooled effect sizes and give greater weight to studies with small samples [34] (Berkeljon & Baldwin, 2009). At the same time, the suitability of the random effect model will be verified by the heterogeneity test. That is, the significance of the Cochran Q test results and the I^2^ value will be checked. If the Q test result is significant or the I^2^ value is higher than 75% (I^2^ 25%, 50%, and 75% represent low, medium, and high heterogeneity, respectively), indicating that there is heterogeneity among studies that cannot be ignored. In this case, it is more appropriate to choose a random-effects model; otherwise, a fixed-effects model should be chosen [35] (Higgins et al., 2003).

## 3. Results

### 3.1. Search Results

After searching multiple databases, a total of 1139 studies were identified. After 605 irrelevant or duplicate records were removed by title, the remaining 534 studies were further evaluated according to the following eligibility criteria: (1) inconsistent subjects, (2) inconsistent study content, non-empirical studies, non-experimental studies, and (3) studies without primary data and full text were excluded. After screening according to the above criteria, a total of 14 valid pieces of literature were obtained, including 0 Chinese literature and 14 English literature (Figure 1).

### 3.2. Description of Included Studies

After literature search and screening, 14 original pieces of literature were finally included, with a total of 14 intervention items, 44 effect sizes, and a total of 7873 students. Among them, there are 14 English literature and 0 Chinese literature; 5 randomized control experimental studies (RCT), 9 non-randomized control experimental studies; 5 studies with follow-up measurement and 10 studies without follow-up measurement; and studies in 4 developing countries, and 11 experimental studies in developed countries (Appendix A
Table A1). 

### 3.3. Outcomes

#### 3.3.1. Publication Bias and Heterogeneity Testing

The heterogeneity test was performed on the studies included in the meta-analysis, and the Cochran Q test results were significant (*p* < 0.001), and I2 > 75%, indicating that the effect values of the 14 original studies included in the meta-analysis had non-negligible heterogeneity. Meta-analyses with random effects models are accurate (Table 2).

Both the immediate effect of the intervention and the delay effect of the intervention were presented at the top of the inverted funnel and were evenly distributed on both sides of the total effect, indicating that there is less possibility of publication bias from a subjective judgment point of view (Figure 2 and Figure 3). The *p* value of Egger’s linear regression coefficient of the immediate effect of the intervention and the delay effect of intervention were not significant (*p* = 0.895, *p* = 0.285), which objectively indicated that there was no publication bias (Table 2).

#### 3.3.2. Main Effects and Sensitivity Tests

The main effects included knowledge, stigma, and help-seeking effects in the immediate effect of the intervention and the follow-up effect of the intervention. In terms of the immediate effect of the intervention, knowledge and stigma were moderately large and moderately small (g_knowledge_ = 0.622, g_stigma_ = 0.262, *p* < 0.001), and the effect of the help-seeking intervention was not significant (g = 0.078, *p* = 0.105). In the intervention’s delay effect, the intervention effects of knowledge, stigma and help-seeking were not significant (*p* > 0.001). In the sensitivity analysis, the combined effect size of knowledge, attitude, and help-seeking effect and delay effect size did not change after removing each effect value, indicating that the immediate effect size and delay effect size of the three were stable (Table 3).

## 4. Discussion

This study draws the following results: (1) the interventions significantly improved mental health knowledge and reduced stigma in the short term but failed to significantly improve the willingness or behavior to seek help; (2) different interventions have different effects on students’ mental health literacy. Judging from the immediate effect of the intervention on students’ mental health literacy knowledge, stigma, and help-seeking, the intervention measures have the most obvious effect on improving students’ mental health knowledge, with a medium-to-large effect size, followed by the improvement of students’ stigma experienced as a result of their mental health. The effect size was moderately small, and the students’ help-seeking willingness or behavior did not improve significantly (*p* = 0.105). However, from the perspective of the delay effect, the intervention measures did not significantly improve the knowledge, stigma, and help-seeking of students’ mental health literacy, which is similar to the previous research results [22,24,36] (Theda Rose, et al., 2017; Yael Perry, et al., 2014; Darcy et al., 2007). It shows that under the current shortage of mental health resources and insufficient teacher-student ratios in schools, relying on the unique advantages of ordinary teachers can undertake the task and responsibility of improving students’ mental health literacy and promoting mental health level and, to a certain extent, relieve the tension of school mental health resources. However, from the perspective of different dimensions of mental health literacy, the effects of knowledge, stigma and seeking help are quite different. From the perspective of the duration of the intervention effect, the intervention delay effect was not significant, indicating that students’ mental health literacy would change over time.

The above results may be due to the following reasons: first, from the perspective of teachers’ classroom teaching methods and content, mainly lectures and discussions, such as written materials, electronic resources, classroom discussions, case analysis, group reflection, etc., and indirect contact with patients through video. The supplementary teaching method is more suitable for improving mental health knowledge and the change of attitude [37] (Robert Milin et al., 2016), so that students can understand the common symptoms of psychological disorders and the way to seek help. However, the actual willingness to seek help or behavior change is more suitable for participatory teaching methods, such as role-playing and practical simulation [22,23] (Theda Rose et al., 2017, Paul B et al., 2009), so that students not only know how to seek help, but more importantly, they are willing to seek help. It takes time to seek help for the actual behavioral process, so it is impossible to obtain obvious effects in the immediate measurement after the experiment; thirdly, the length and intensity of the intervention measures are insufficient. An individual’s mental health is a dynamic process of change, which will change with personal experience, the degree of pain, the surrounding environment, and emergencies. Therefore, the intervention in students’ mental health literacy should run through the students’ entire study life. In terms of the duration of intervention in the characteristics of the original research, the duration of intervention in the original research included in the meta-analysis was mostly 1–2 months. Only short-term intervention cannot meet the sustainable development of students’ mental health literacy.

## 5. Conclusions

In conclusion, the results of this systematic review and meta-analysis suggest that general classroom teachers can effectively improve students’ mental health literacy, especially their mental health knowledge and stigma. In future, interventions should be expanded to cover the entire student life, with specific interventions selected based on the age and grade of the student. For example, randomized controlled studies should be used wherever possible to prevent selection bias and the influence of external circumstances on experimental results. Different intervention methods are adopted for different dimensions of mental health literacy, such as improving students’ mental health knowledge through lectures and guiding students to resonate with people with mental disorders through contact (on-site contact and video contact), thereby reducing their stigmatization of mental disorders. Finally, the aim should be to strengthen mental health education and training for all teachers, including normal university students and in-service teachers. Mental health education courses will be included in the compulsory courses for normal university students. In-service teachers should receive regular mental health education training, which should be included in their daily training plans.

## Figures and Tables

**Figure 1 ijerph-20-00949-f001:**
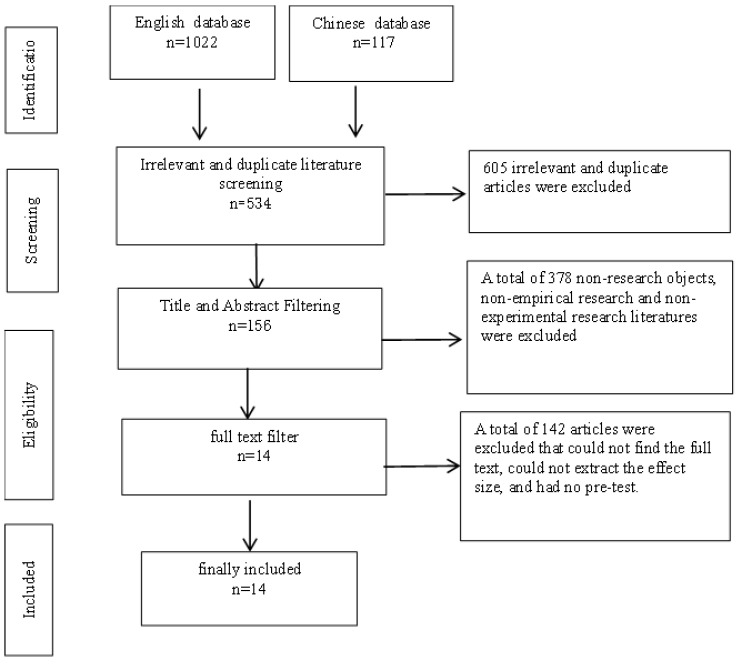
The flow chart of literature screening.

**Figure 2 ijerph-20-00949-f002:**
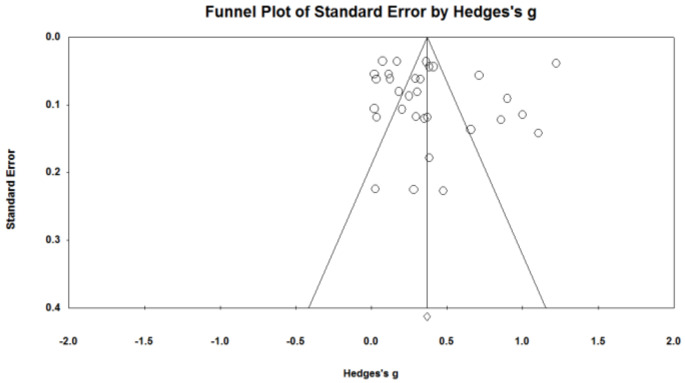
Instantaneous effect volume funnel diagram.

**Figure 3 ijerph-20-00949-f003:**
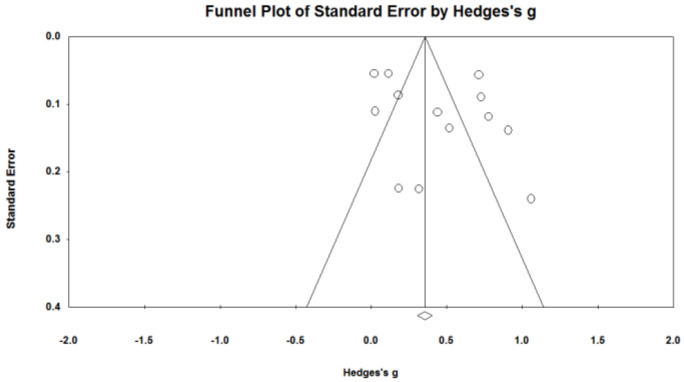
Delay effect Funnel Chart.

**Table 1 ijerph-20-00949-t001:** Code table.

Variable	Coding
outcome variable	Knowledge = 1; stigma = 2; help-seeking = 3

**Table 2 ijerph-20-00949-t002:** Heterogeneity test and publication bias Egger’s linear regression.

Outcome Variable	k	Publication Bias Test				Heterogeneity Test			
		Egger’s Intercept	SE	95% CI	*p*	Q-Value	df	*p*	I^2^
immediate effect of intervention	31	0.280	2.108	(−4.031, 4.591)	0.895	867.482	30	0.000	96.542
Intervention delay effect	13	2.621	2.332	(−2.513, 7.756)	0.285	166.027	12	0.000	92.772

**Table 3 ijerph-20-00949-t003:** Main effects and sensitivity tests.

Outcome Variable		k	g (95% CI)	Sensitivity Test	Heterogeneity Test			
				g (95% CI)	Q_w_	df	*p*	I^2^
Knowledge	immediate effect of intervention	12	0.622 (0.395, 0.849)	0.622 (0.395, 0.849)	396.399	11	0.000	97.225
	Intervention delay effect	5	0.752 (0.671, 0.834)	0.752 (0.671, 0.834)	3.480	4	0.481	0.000
stigma	immediate effect of intervention	14	0.262 (0.170, 0.354)	0.262 (0.170, 0.354)	79.760	13	0.000	83.701
	Intervention delay effect	5	0.288 (0.123, 0.452)	0.288 (0.123, 0.452)	12.648	4	0.013	68.374
Help-seeking	immediate effect of intervention	5	0.078 (−0.033, 0.189)	0.078 (−0.033, 0.189)	7.662	4	0.105	47.796
	Intervention delay effect	3	0.029 (−0.065, 0.123)	0.029 (−0.065, 0.123)	0.497	2	0.780	0.000

## Data Availability

Data sharing is not applicable to this article.

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
