# Peer review of "Research on the Effect of Evidence-Based Intervention on Improving Students’ Mental Health Literacy Led by Ordinary Teachers: A Meta-Analysis"

_ijerph, 2023, doi:10.3390/ijerph20020949_

Round 1

Reviewer 1 Report

This topic is important. However there are major concerns on every section:

1.      Introduction

(1)   Some references are too old. E.g. line 34.

(2)   The rationale for reasons for insufficient services that the adolescents received is not clearly addressed. (line 43-47.)

(3)   Many references were not provided in line 66-69.

(4)   The definition of ordinary teachers and the importance to study interventions led by ordinary teachers should be provided.

(5)   In line 73, what kinds of differences of experimental results should be described.

(6)   The definition, dimensions, and measurement of mental health literacy should be reviewed in this section.

2.      Methods

(1)2.1: The search terms did not show how to identify intervention led by ordinary teachers.

(2)2.2: ‘Three or one’ in line 101 sounds strange.

(3)2.2: What does ‘research object’ mean in this study? (line 105)

(4)2.3: The information is too few to draw a table.

3.      Results:

(1)3.2: The total number of included studies is 14. However, the total number seems to be 15. (line 181-183)

(2)3.3: In line 189, the meaning of ‘Meta-analysis …….are accurate’ is not clear.

(3)In Table A1: What does classroom teaching mean exactly? How to differentiate this to other teaching method?

4.      Discussion:

(1)   line 216: ‘boosting effect’ à This sentence is not clear.

(2)   It may be better to replace ‘tracking effect’ with ‘delay effect’.

(3)   Line 237-240: It says ‘supplementary teaching method is more suitable….’. and ‘participatory teaching methods ….. These statements should cite proper references.

5.      Conclusion:

(1)   Is ‘general classroom teachers’ equal to ‘ordinary teacher’?

(2)   Line 257: What does ‘the entire student life’ mean?

(3)   Line 264: reducing their awareness or increasing?

(4)   Line 265: The last sentence is not clear.

6.      More reference (ref.) are expected.

(1)   line 43: in developing countries (ref.???)

(2)    

7.      Grammar problems

(1) Missing punctuation marks : E.g. line 37, line 237 (video), line 249

   (2) incomplete sentences (IS): Line 258

   (3) inconsistence between subjects and verbs: E.g. line 94

   (4) tense: E.g. line 114 (‘will be’ should be ‘was’)2.5.2: In this paragraph, some ‘will be’ should be ;’was’.

   (5) extra word(s): line 229 (condition.), line 141-142 (stigma and stigma)

Reviewer 2 Report

This is an interesting study - enhancing mental health literacy in school classrooms is important.

There are a number of minor, yet important, grammatical errors to be addressed to strengthen expression and readability.

Title and Abstract: the word 'ordinary', as in ordinary teachers is unnecessary and raises more questions than it answers - how is ordinariness being assessed and measured? Is there a comparison group of non-ordinary teachers? What demographics or training or skills distinguish the groups?

Within abstract (page 1 line 12) the sentence 'Methods: A systematic search using 5 English (Web of science、PubMed、ProQuest、EBSCO、Springer Link) and 3 Chinese (CNKI, WanFang, and VIP) databases were initiated...' it should be 'was' initiated (this relates to the search - singular - that was initiated).

Introduction: page 2 line 46, there is no full stop before the brackets with citations at the end of the sentence). There needs to be a space before a bracket (page 2 line 51 and 60), and a space before a new sentence (page 2 line 52 and 54). Re-word line 70 - 'let students contact with mentally ill patients' - it is not accurate that viewing a video provides such contact and it would be preferable to use the language of mental health rather than illness, especially in a paper concerned with mental health literacy.

Materials and Methods: Watch tense - page 3 line 113, 'Discrepancies are discussed until consensus is reached, and any disagreements regarding inclusion will be discussed and resolved...' should be 'were' discussed and 'was' reached; this is in the past.

Results: 3.2 Description of Included Studies page 5 seems to include 15 rather than 14 studies?

Discussion: page 7 line 220 'improvement of students' psychological barrier stigma' is unclear in meaning. Page 7 line 243, there is a semi-colon where it should be a full stop. 

Line 246-8 requires references/citations to substantiate: 'An individual's mental health is a dynamic process of change, which will change with personal experience, the degree of pain, the surrounding environment, and emergencies.'
